# The deformed Inozemtsev spin chain

**Rob Klabbers**[1][⋆]**, and Jules Lamers**[2]⌢[3][†]

**1** Humboldt-Universität zu Berlin
Zum Großen Windkanal 2, 12489 Berlin, Germany

**2** Université Paris–Saclay, CNRS, CEA
Institut de Physique Théorique
91191 Gif-sur-Yvette, France

**3** Deutsches Elektronen-Synchrotron DESY
Notkestraße 85, 22607 Hamburg, Germany

⋆ rob.klabbers@physik.hu-berlin.de , † jules.lamers@desy.de

## Abstract

**The Inozemtsev chain is an exactly solvable interpolation between the short-range Hei-senberg and long-range Haldane–Shastry (HS) chains. In order to unlock its potential to study spin interactions with tunable interaction range using the powerful tools of integrability, the model's mathematical properties require better understanding. As a major step in this direction, we present a new generalisation of the Inozemtsev chain with spin symmetry reduced to $U(1)$, interpolating between a Heisenberg xxz chain and the xxz-type HS chain, and integrable throughout. Underlying it is a new quantum many-body system that extends the elliptic Ruijsenaars system by including spins, contains the trigonometric spin-Ruijsenaars–Macdonald system as a special case, and yields our spin chain by 'freezing'. Our models have potential applications from condensed-matter to high-energy theory, and provide a crucial step towards a general theory for long-range integrability.**

## 1  Introduction

Recent years brought tremendous progress for trapped-ion and cold-atom experiments, and low-dimensional systems with tunable spin-spin interactions can now be engineered [1–4]. Wheareas such systems inherently have *long-range* spin interactions, theoretical studies often assume drastically simplified nearest-neighbour interactions. Long-range spin interactions also find applications in quantum information and computing [5–7] and pose fundamental questions about e.g. causality [8–11]. In $1 + 1$ dimensions, *(quantum) integrable* models are exactly solvable thanks to underlying symmetries. Such models may thus offer exciting opportunities to study the effects of long-range interactions using exact analytical methods. Yet such models are rare, and the theory behind them is incomplete.

**Main results.**   We introduce two new integrable long-range models with spins:

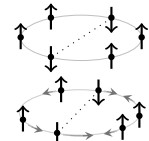

a (quantum) spin chain;

a quantum many-body system (QMBS),
of particles with spins *moving* on a circle.

As we shall see, the two models are closely related. Besides having potential applications in both condensed-matter and high-energy theory, our models shed light on the three-decade old open problem to understand the integrability of the Inozemtsev chain.

**The spin chain.**   Until recently, the study of integrable long-range spin chains focused on *isotropic* (i.e. $SU(2)$-symmetric) models, with hamiltonian of the form

$$H^{\text{iso}} = \frac{1}{2}\sum_{i<j}^{N} \bar{V}(i-j)\big(1 - \vec{\sigma}_i \cdot \vec{\sigma}_j\big) = \sum_{i<j}^{N} \bar{V}(i-j)\big(1 - P_{ij}\big), \tag{1}$$

where we consider a chain of $N$ spins, $\bar{V}(x)$ is a pair potential setting the interaction range, $\vec{\sigma} = (\sigma^x, \sigma^y, \sigma^z)$ are the Pauli spin matrices, and $P_{ij} = (1 + \vec{\sigma}_i \cdot \vec{\sigma}_j)/2$ is the spin permutation operator. The Haldane–Shastry (HS) chain [12,13] is given by (1) with pair potential

$$\bar{V}^{\text{HS}}(x) = \frac{1}{r^2}, \quad r = \frac{N}{\pi}\sin\big|\tfrac{\pi}{N}x\big|, \tag{2}$$

which is the critical case for long-range order (cf. [8,10,14]). It can be engineered with trapped ions [15] and is a lattice toy model for the fractional quantum Hall effect [16,17] and Wess–Zumino–Witten CFT [18–21]. This model is connected (Fig. 1) to the nearest-neighbour Heisenberg xxx chain through the Inozemtsev chain [22], whose hamiltonian $H^{\text{Ino}}$ is given by (1) with

$$\bar{V}^{\text{Ino}}(x) = \wp(x) + \text{cst} \tag{3}$$

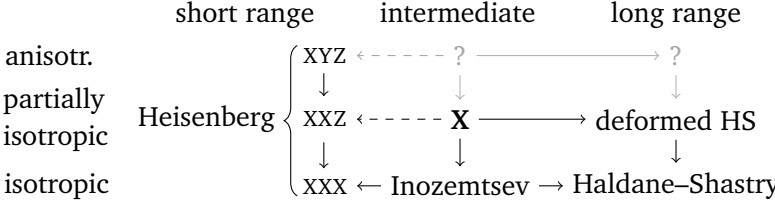

Figure 1: Landscape of integrable long-range spin chains, including the Heisenberg and Haldane–Shastry chains and their partially isotropic extensions. We find the spot marked '**X**'.

the Weierstraß elliptic function. This pair potential generalises (2) by including a second, imaginary period that sets the interaction range. Widely believed to be integrable [17,23], $H^{\text{Ino}}$ offers the tantalising possibility to study a spin system analytically as one tunes the interaction range. First, however, the toolkit of integrability needs to be developed further: there is a conjecture for the conserved charges of $H^{\text{Ino}}$ [24,25], but no underlying algebraic structure is known. This is an important open problem in the theory of integrability [17]. To unveil such structures we shall break the spin symmetry of $H^{\text{Ino}}$ in a controlled way.

The HS chain has a *partially isotropic* (i.e. $U(1)$-symmetric) extension retaining its key properties, the *deformed* HS chain [26–28]. Our first new long-range model likewise deforms $H^{\text{Ino}}$, generalising the Inozemtsev and deformed HS chains as in Fig. 1 while remaining integrable. The partially isotropic generalisation of $1-P_{ij}$ from (1) comes in two 'chiralities', with *deformed permutations* transporting either spin to the other, for a *deformed exchange,* followed by transport back. Like in (1), a *potential* sets the interaction range; it is a 'point splitting' of (3) as anticipated in [23].

**The QMBS.** Unlike for nearest-neighbour models, integrability of long-range spin chains hinges on connections to QMBSs of Calogero–Sutherland (CS) and Ruijsenaars type. This is best understood for HS (see also [29]):

    i. its exact wavefunctions come from a *spinless* trigonometric CS system [16,30],

    ii. its conserved charges stem from a trigonometric CS system *with spins* by 'freezing' [30–32],

and the enhanced (Yangian) spin symmetry of $H^{\text{HS}}$ arises from (ii) too [30,33]. These connections persist at the partially isotropic level, where trigonometric CS is generalised to the 'relativistic' trigonometric Ruijsenaars–Macdonald (RM) system [26,28,30] (Fig. 2). For $H^{\text{Ino}}$ only (i) was properly understood, via the *elliptic* CS system [23,34]. Here, we add (ii): our spin chain arises by freezing an *elliptic dynamical spin-Ruijsenaars system*. This QMBS is our second new long-range model (Fig. 2). Despite its supporting role here, it is clearly of independent theoretical interest. We shall prove the commutativity of its hamiltonians elsewhere.

**Outline.** While we focus on spin 1/2, all our results extend to multi-component versions with several particle 'species' ('colours').[1] In Section 2, we introduce our new long-range spin chain, discuss how it satisfies the defining properties introduced above, and compute two new limits: an intermediate refinement of the Inozemtsev chain, and the short-range limit. We furthermore point out some interesting new features. In Section 3, we construct a novel QMBS and discuss its properties. We moreover outline how 'freezing' this QMBS yields our

---

[1] Simply replace (8) by the dynamical $\mathfrak{gl}_r$ $R$-matrix [35].

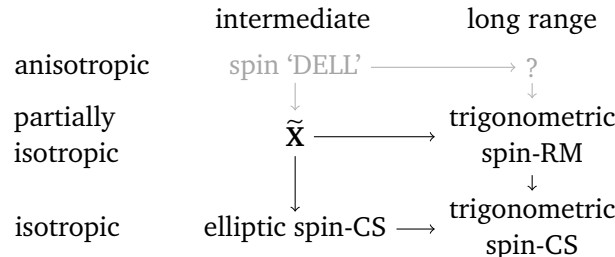

Figure 2: Landscape of integrable QMBS with spins, including Calogero–Sutherland (CS) and Ruijsenaars–Macdonald (RM). Without lattice spacing as an infrared cutoff, the short-range limit is absent. We find the spot marked '$\widetilde{\mathbf{X}}$'.

spin chain, thereby connecting the commutativity of their respective charges and hence their integrability. We conclude in Section 4. The appendices contain all relevant information about the elliptic functions (Appendix A) and $R$-matrix (Appendices B–C) that we will need.

## 2 The spin chain

### 2.1 Hamiltonians

Consider $N$ spin-1/2 sites equispaced on a circle. The deformed Inozemtsev chain has 'chiral' hamiltonians

$$H^{\mathrm{L}} = \sum_{i<j}^{N} V(i-j)\, S^{\mathrm{L}}_{[i,j]}, \quad H^{\mathrm{R}} = \sum_{i<j}^{N} V(i-j)\, S^{\mathrm{R}}_{[i,j]}. \tag{4}$$

Let $\rho(x) = \theta'(x)/\theta(x)$, where $\theta(x)$ is the odd Jacobi theta function with quasiperiods $\mathrm{i}\pi/\kappa$ and $N$, which we view as a periodisation of a hyperbolic sine:

$$\theta(x) = \frac{\sinh(\kappa x)}{\kappa} \prod_{n=1}^{\infty} \frac{\sinh[\kappa\,(N\,n+x)]\,\sinh[\kappa\,(N\,n-x)]}{\sinh^2(N\kappa\,n)} = \frac{\sinh(\kappa x)}{\kappa} + O(p^2), \tag{5}$$

with nome $p = \mathrm{e}^{-N\kappa}$, see Appendix A for more.

The potential is

$$V(x) = \frac{\rho(x-\eta) - \rho(x+\eta)}{\theta(2\eta)} \sim \frac{1}{\mathrm{sn}(x+\eta)\,\mathrm{sn}(x-\eta)}, \tag{6}$$

with anisotropy parameter $\eta$. Here sn is the Jacobi elliptic sine function, see (A.6).

The long-range spin interactions $S^{\mathrm{L}}_{[i,j]}$ and $S^{\mathrm{R}}_{[i,j]}$ are deformations of the isotropic long-range spin exchange interaction $E_{ij} = (1 - P_{ij})/2 = (1 - \vec{\sigma}_i \cdot \vec{\sigma}_j)/4$ in (1). The latter admits two 'chiral' decompositions into nearest-neighbour steps:

$$\begin{aligned} E_{ij} &= P_{j-1,j} \cdots P_{i+1,i+2}\, E_{i,i+1}\, P_{i+1,i+2} \cdots P_{j-1,j} \\ &= P_{i,i+1} \cdots P_{j-2,j-1}\, E_{j-1,j}\, P_{j-2,j-1} \cdots P_{i,i+1}. \end{aligned} \tag{7}$$

The structure on the right-hand side persists to the partially isotropic level, with suitable replacements for both the spin permutation $P$ and the nearest-neighbour spin interaction $E$. These are both built from Felder's dynamical $R$-matrix [36]

$$\check{R}(x,a) = \begin{pmatrix} 1 & 0 & 0 & 0 \\ 0 & f(\eta,x,\eta\,a) & f(x,\eta,\eta\,a) & 0 \\ 0 & f(x,\eta,-\eta\,a) & f(\eta,x,-\eta\,a) & 0 \\ 0 & 0 & 0 & 1 \end{pmatrix}, \quad f(x,y,z) = \frac{\theta(x)\,\theta(y+z)}{\theta(x+y)\,\theta(z)}, \tag{8}$$

depending on a 'dynamical' parameter $a$. It satisfies the dynamical Yang–Baxter equation, see Appendix B. The deformed spin permutation is

$$P_{i,i+1}(x) = \check{R}_{i,i+1}\big(x, a - (\sigma_1^z + \cdots + \sigma_{i-1}^z)\big) = a \uparrow \cdots \uparrow \overset{x'' \; x'}{\underset{x' \; x''}{\bigtimes}} \uparrow \cdots \uparrow, \quad x = x' - x'', \qquad (9)$$

where the $i$th and $i + 1$st spins cross, carrying along their 'inhomogeneity' parameters $x', x''$. The dynamical parameter $a$ is shifted by the spin-$z$ to the left of the $R$-matrix. On the usual spin basis labelled by $s_j$ equal to $\uparrow \equiv +1$ or $\downarrow \equiv -1$ for each $1 \leqslant j \leqslant N$ this means

$$P_{i,i+1}(x)|s_1, \ldots, s_N\rangle = |s_1, \ldots, s_{i-1}\rangle$$
$$\otimes \check{R}\big(x, a - \textstyle\sum_{k=1}^{i-1} s_k\big)|s_i, s_{i+1}\rangle$$
$$\otimes |s_{i+1}, \ldots, s_N\rangle,$$

so, for example, $P_{23}(x) = |\uparrow\rangle\langle\uparrow| \otimes \check{R}(x, a - 1) + |\downarrow\rangle\langle\downarrow| \otimes \check{R}(x, a + 1)$. The properties of these deformed spin permutations are collected in Appendix B.

Finally, the deformed nearest-neighbour spin exchange is defined from (9) as

$$E_{i,i+1}(x) = \frac{1}{\theta(\eta) V(x)} P_{i,i+1}(-x) P'_{i,i+1}(x) = a \uparrow \cdots \uparrow \overset{x' \; x''}{\underset{x' \; x''}{\bowtie}} \uparrow \cdots \uparrow, \quad x = x' - x''. \qquad (10)$$

This definition, in which we factor out the potential (6), is chosen such that both (6) and (10) have the appropriate limits, as we will see in Section 2.2. The explicit $4 \times 4$ matrix determining (10) is given in Appendix C. Unlike the potential, it depends on $a$. While the dependence on $x$ is new compared to the Inozemtsev and deformed HS chains, this feature is shared by the elliptic long-range spin chain of Matushko and Zotov [37, 38], as well as in all degenerations thereof.

Together, the deformed permutation (9) and deformed exchange (10) define the chiral long-range spin interactions $S^{\mathrm{L}}_{[i,j]}$ and $S^{\mathrm{R}}_{[i,j]}$ diagrammatically as

Here each site $1 \leqslant k \leqslant N$ has a fixed inhomogeneity parameter $x_k^\star = k$. Thus, the deformed long-range spin interactions (11) read

$$S^{\mathrm{L}}_{[i,j]} = P_{j-1,j}(1) \cdots P_{i+1,i+2}(j-i-1) E_{i,i+1}(i-j) P_{i+1,i+2}(i-j+1) \cdots P_{j-1,j}(-1), \qquad (12)$$

$$S^{\mathrm{R}}_{[i,j]} = P_{i,i+1}(1) \cdots P_{j-2,j-1}(j-i-1) E_{j-1,j}(i-j) P_{j-2,j-1}(i-j+1) \cdots P_{i,i+1}(-1), \qquad (13)$$

in clear analogy to the first and second line, respectively, of the decompositions in (7).

**Examples.** At $N = 3$ the chiral long-range spin interactions read

$$\begin{aligned}
S^{\mathrm{L}}_{[1,2]} &= E_{12}(-1), & S^{\mathrm{L}}_{[2,3]} &= E_{23}(-1), & S^{\mathrm{L}}_{[1,3]} &= P_{23}(1) E_{12}(-2) P_{23}(-1), \\
S^{\mathrm{R}}_{[1,2]} &= E_{12}(-1), & S^{\mathrm{R}}_{[2,3]} &= E_{23}(-1), & S^{\mathrm{R}}_{[1,3]} &= P_{12}(1) E_{23}(-2) P_{12}(-1).
\end{aligned} \qquad (14)$$

For higher $N$ the first few terms look exactly the same, with dependence on $N$ residing in the real (quasi)period of the entries of $E_{i,i+1}(x)$ and $P_{i,i+1}(x)$. At $N = 4$ we further need

$$
S^{\mathrm{L}}_{[3,4]} = E_{34}(-1), \quad S^{\mathrm{L}}_{[2,4]} = P_{34}(1) E_{23}(-2) P_{34}(-1),
$$
$$
S^{\mathrm{L}}_{[1,4]} = P_{34}(1) P_{23}(2) E_{12}(-3) P_{23}(-2) P_{34}(-1),
\tag{15a}
$$

for the left-chiral interactions, and for the right-chiral ones

$$
S^{\mathrm{R}}_{[3,4]} = E_{34}(-1), \quad S^{\mathrm{R}}_{[2,4]} = P_{23}(1) E_{34}(-2) P_{23}(-1),
$$
$$
S^{\mathrm{R}}_{[1,4]} = P_{12}(1) P_{23}(2) E_{34}(-3) P_{23}(-2) P_{12}(-1).
\tag{15b}
$$

In general one obtains $S^{\mathrm{L}}_{[i,j]}$ from $S^{\mathrm{L}}_{[1,j-i+1]}$ by shifting all subscripts $k, k+1$ to $k+i-1, k+i$. The same holds for $S^{\mathrm{R}}_{[i,j]}$. Note that the $S^{\mathrm{L}}_{[i,j]}$ have the same structure as in [27] and the $S^{\mathrm{R}}_{[i,j]}$ look like in [28], the difference being the choice of $R$-matrix.

## 2.2  Properties and limits

While the hamiltonians (4) are more complex than in the isotropic case (1), their ingredients have clear physical meanings: a potential (6), a deformed permutation (9), and a deformed spin exchange (10). There are four parameters: the length $N \geqslant 2$, $\kappa > 0$ tuning the interaction range, the anisotropy $\eta$, and the dynamical parameter $a$.[2] The spectrum is real if $\eta$ is imaginary (i.e. the regime $|\Delta| > 1$ for the Heisenberg XXZ spin chain) and $a$ real.

**Defining properties.**   The chain (4) contains the Inozemtsev and deformed HS chains as in Fig. 1, and is integrable. Let us explain.

When $\eta \to 0$ we retrieve the isotropic Inozemtsev hamiltonian $H^{\mathrm{Ino}}$ given by (1) and (3). Indeed, (6) becomes $-\rho'(x) = \bar{V}^{\mathrm{Ino}}(x)$ from (3), and both (12)–(13) yield $1 - P_{ij}$ up to a conjugation that is removed by $a \to -\mathrm{i}\infty$, since then $P_{i,i+1}(x) \to P_{i,i+1}$ and $E_{i,i+1}(x) \to 1 - P_{i,i+1}$.

At $\kappa = 0$ we find the deformed HS chain, again up to a conjugation that disappears if $a$ is removed. The potential (6) has the long-range limit $V^{\mathrm{tri}}(x) = (\frac{\pi}{N})^2 / \sin[\frac{\pi}{N}(x+\eta)] \sin[\frac{\pi}{N}(x-\eta)]$. If moreover $\eta a \to -\mathrm{i}\infty$, the exchange (10) becomes independent of $x$, namely

$$
E^{\mathrm{tri}} = \begin{pmatrix} 0 & 0 & 0 & 0 \\ 0 & q^{-1} & -q & 0 \\ 0 & -q^{-1} & q & 0 \\ 0 & 0 & 0 & 0 \end{pmatrix}, \quad q = \mathrm{e}^{\pi \mathrm{i} \eta / N},
\tag{16}
$$

acting at sites $i, i+1$. The deformed permutation (9) reduces to the operator

$$
\check{R}^{\mathrm{tri}}(x) = 1 - \frac{\sin(\frac{\pi}{N} x)}{\sin[\frac{\pi}{N}(x+\eta)]} E^{\mathrm{tri}}
\tag{17}
$$

at sites $i, i+1$. We will discuss the algebraic meaning of (16)–(17) in Section 2.3. Thus we obtain the deformed HS chain, which is still chiral and of the form (4). Further letting $\eta \to 0$, both reduce to the isotropic HS hamiltonian $H^{\mathrm{HS}}$, which is also obtained from $H^{\mathrm{Ino}}$ as $\kappa \to 0$.

Finally, our model is integrable in the sense that the chiral hamiltonians (4) commute,

$$
[H^{\mathrm{L}}, H^{\mathrm{R}}] = 0,
\tag{18}
$$

belonging to a tower of conserved charges whose expressions parallel those in [28,37], see [39].

---

[2] These parameters have some constraints, since the potential (6) has poles at $2\eta = N k + \mathrm{i}\pi l / \kappa$ for $k, l \in \mathbb{Z}$, and the entries of (8) have poles at $\eta a = N k + \mathrm{i}\pi l / \kappa$.

 **Further properties.** The ordinary Inozemtsev chain has full $SU(2)$ spin symmetry. Our chain
is its generalisation with spin symmetry broken to $U(1)$: our conserved charges all commute
with $S^z = \sum_i \sigma_i^z/2$.

Like the deformed HS chain, the spin interactions (11) involve multispin interactions af-
fecting all intermediate spins, whence the subscript '$[i, j]$'. While $\eta \neq 0$ breaks periodicity, our
chain has quasiperiodic boundary conditions. One of the conserved charges is the deformed
(lattice) translation operator (cf. [27])

$$G = a \quad \begin{matrix} 2 & \cdots & N & 1 \\ & & & \\ 1 & 2 & \cdots & N \end{matrix} \quad = K_N \, P_{N-1,N}(1-N) \cdots P_{12}(-1), \quad K_N = \mathrm{e}^{-\kappa \eta [a - (\sigma_1^z + \cdots + \sigma_{N-1}^z)] \sigma_N^z}. \tag{19}$$

Here $K_N$ is a diagonal twist, $\mathrm{e}^{-\kappa \eta a \sigma^z} = \mathrm{diag}(\mathrm{e}^{-\kappa \eta a}, \mathrm{e}^{\kappa \eta a})$, acting at site $N$ with a shift of $a$ as
in (9). Upon normalisation, (2.2) provides a notion of momentum, plus all $N$ eigenvectors
at $S^z = N/2 - 1$ (cf. §1.2.6 in [28]), i.e. the magnons of our chain. We have not yet been
able to find an expression for the dispersion relation. Moreover, (2.2) allows us to express the
long-range interaction of neighbouring spins on sites 1 and $N$ as

$$S_{[1,N]}^{\mathrm{L}} = G \, S_{[1,2]}^{\mathrm{L}} \, G^{-1}, \quad S_{[1,N]}^{\mathrm{R}} = G^{-1} S_{[N-1,N]}^{\mathrm{R}} \, G, \tag{20}$$

underlining the chirality of the hamiltonians (4).

**New limits.** Our chain has various new limits. For $N \to \infty$ we formally get a hyperbolic
counterpart of the deformed HS chain, with $N \leftrightsquigarrow \mathrm{i}\pi/\kappa$ and sum in (4) over all integers.
Numerics suggests that its matrix entries converge.

As discussed in the previous section, the limit $\eta \to 0$ yields the Inozemtsev spin chain (up
to a conjugation). Interestingly, this limit can be refined to obtain an intermediate spin chain
that seems to be new, by setting $a = a'/\eta$ before sending $\eta \to 0$. This does not affect the
limits of the potential and deformed spin permutation, but changes the limit of the deformed
exchange (10) as a function of $a'$. Both chiral hamiltonians (4) then limit to

$$H^{\mathrm{Ino}}(a') = \frac{1}{2} \sum_{i<j}^{N} \left( \phi'(i-j, a') \frac{\sigma_i^+ \sigma_j^-}{2} + \phi'(i-j, -a') \frac{\sigma_i^- \sigma_j^+}{2} + \bar{V}^{\mathrm{Ino}}(i-j) \, 1 - \sigma_i^z \sigma_j^z \right), \tag{21}$$

where $\phi'$ is the derivative with respect to the first variable of $\phi(x, y) = \theta(x+y)/[\theta(x)\theta(y)]$.
The hamiltonian (21) generalises $H^{\mathrm{Ino}}$ from (1) and (3) with an extra parameter $a'$ that breaks
the left-right symmetry and $SU(2)$ spin symmetry. Unlike for $\eta \neq 0$, (21) is not dynamical in
the sense that the parameter $a'$ does not receive any shifts as in e.g. (9). The spectrum is
$a'$-dependent and real when $a' \in \mathrm{i}\mathbb{R}$. The isotropic Inozemtsev chain is retrieved by sending
$a' \to 0$ or $a' \to \mathrm{i}\pi/\kappa$, since then $\phi'(x, a') \to \rho(x) = -\bar{V}^{\mathrm{Ino}}(x)$.

Finally, we turn to the short-range limit $\kappa \to \infty$. It is convenient to represent the potential
(6) as the sum

$$\begin{aligned} \rho(x+\eta) - \rho(x-\eta) &= \sum_{n \in \mathbb{Z}} \frac{2\kappa \sinh(2\kappa\eta)}{\sinh[\kappa(\eta+x+Nn)] \sinh[\kappa(\eta-x-Nn)]} \\ &= \sum_{n \in \mathbb{Z}} \frac{4\kappa \sinh(2\kappa\eta)}{\cosh(2\kappa\eta) - \cosh[2\kappa(Nn+x)]}. \end{aligned} \tag{22}$$

For a convergent but non-zero limit as $\kappa \to \infty$ we must also send $\eta \to 0$ with $\kappa\eta$ fixed so
that $\cosh(2\kappa\eta)$ becomes constant. Thus we set $\eta = -\mathrm{i}\pi\gamma/\kappa$ and rescale (22) by a prefactor

behaving as $n_\eta(\kappa) \sim \mathrm{e}^{2\kappa}/[4\kappa \sinh(2\kappa\eta)]$ to obtain

$$n_{-\mathrm{i}\pi\gamma/\kappa}(\kappa)\big(\rho(x-\mathrm{i}\pi\gamma/\kappa)-\rho(x+\mathrm{i}\pi\gamma/\kappa)\big) \to \delta_{x,1}+\delta_{x,N-1}, \quad \kappa \to \infty, \quad x \in \{1,\dots,N-1\}. \tag{23}$$

A choice of normalisation that fits with all other limits is to rescale the hamiltonians (4) by $n_\eta(\kappa) = \sinh^2\kappa/[\kappa^2\,\theta(2\eta)]$. This is why we choose denominator $\theta(2\eta)$ in the potential (A.6) rather than the $2\eta$ from [23]; when $\eta \to 0$ the two have the same behaviour. Therefore, as $\kappa \to \infty$, we get a nearest-neighbour chain

$$H^{\mathrm{XXZ}} = \sum_{i=1}^{N-1} S^{\mathrm{H}}_{[i,i+1]} + S^{\mathrm{H}}_{[1,N]}. \tag{24}$$

Here, the exchange $S^{\mathrm{H}}_{[i,i+1]} = E^{\mathrm{H}}_{i,i+1}\big(a-(\sigma^z_1+\cdots+\sigma^z_{i-1})\big)$ is defined like in (9) in terms of a generalisation of (16):

$$E^{\mathrm{H}}(a) = \begin{pmatrix} 0 & 0 & 0 & 0 \\ 0 & \dfrac{\sin[\pi\gamma(a-1)]}{\sin[\pi\gamma a]} & -\dfrac{\sin[\pi\gamma(a+1)]}{\sin[\pi\gamma a]} & 0 \\ 0 & -\dfrac{\sin[\pi\gamma(a-1)]}{\sin[\pi\gamma a]} & \dfrac{\sin[\pi\gamma(a+1)]}{\sin[\pi\gamma a]} & 0 \\ 0 & 0 & 0 & 0 \end{pmatrix}. \tag{25}$$

Since the two expressions in (20) coincide, the boundary term in (24) admits two forms

$$S^{\mathrm{H}}_{[1,N]} = G^{\mathrm{H}} S^{\mathrm{H}}_{[1,2]} G^{\mathrm{H}-1} = G^{\mathrm{H}-1} S^{\mathrm{H}}_{[N-1,N]} G^{\mathrm{H}}, \tag{26}$$

where (2.2) becomes $G^{\mathrm{H}} = K^{\mathrm{H}}_N P^{\mathrm{H}}_{N-1,N}\cdots P^{\mathrm{H}}_{12}$, with twist $\mathrm{e}^{\mathrm{i}\pi\gamma a\sigma^z}$ and permutation built from $\check{R}^{\mathrm{H}}(a) = 1 - \mathrm{e}^{-\mathrm{i}\pi\gamma}E^{\mathrm{H}}(a)$ as in (9). Note that the arguments $x$ have completely disappeared. This $R$-matrix also appeared in a slightly different form in [40], see (5.28) therein.

The short-range limit (24) is a 'dynamical' variant of the Heisenberg XXZ chain. It is no longer chiral, but remains quasiperiodic, since the twist in (26) prevents removing $a$. When $\gamma \to 0$ we obtain, once more up to a conjugation that vanishes as $a \to -\mathrm{i}\infty$, the usual periodic Heisenberg XXX chain (Fig. 1).

## 2.3  Discussion

**Form of spin interactions.**   The long-range interactions (11) are very specific generalisations of $1 - P_{ij}$. The need for such involved interactions is more clear for the deformed HS chain, so as to maintain the HS chain's integrability, enhanced spin symmetry, and extremely simple exact spectrum [27,28]. In turn generalising the deformed HS chain, our spin chain must have similar spin interactions.

**Choice of $R$-matrix.**   The deformed HS chain already uses an $R$-matrix in its deformed permutations, viz. (17). Its enhanced spin symmetry requires [30,33] $\check{R}^{\mathrm{tri}}$ to be related (by 'Baxterisation') to the Hecke algebra—and, for spin 1/2, the Temperley–Lieb algebra, see (28) below. This necessarily leads to some asymmetry ($P\check{R}P \neq \check{R}$) as in (16). Now, at the partially isotropic level, an elliptic potential asks for an $R$-matrix with elliptic functions, cf. (**??**). The standard choices are

- Baxter's eight-vertex (XYZ) $R$-matrix: $P\check{R}^{8\mathrm{v}}P = \check{R}^{8\mathrm{v}}$, which generalises the symmetric six-vertex (XXZ) $R$-matrix;

- Felder's elliptic dynamical $R$-matrix (8) [36]: $S^z$-symmetric, which generalises the $R$-matrix (17).

They are related by a ('face-vertex') transformation [41],

$$\check{R}^{8v}(x_i - x_j)\, T(x_i, x_j, a) = T(x_j, x_i, a)\, \check{R}(x_i - x_j, a)\,. \tag{27}$$

One might expect the corresponding spin chains to be equivalent. Yet the resulting deformed exchanged interactions (10), containing a derivative in $x$, are not related by the $x$-dependent transformation (27). It appears impossible to obtain (17) from $\check{R}^{8v}$ without (27).[3] Hence our spin chain *differs* from the (fully) anisotropic chain recently found by Matushko and Zotov using $\check{R}^{8v}$ [43], which belongs to a landscape disjoint from Fig. 1 [39]. See [38] for a detailed analysis of this fact.

**Modular family.** As we will see below, 'freezing' in fact produces an $SL(2, \mathbb{Z})$-family of integrable longe-range spin chains. Only two of these have a real spectrum for some parameter range, of which only (4) has a short-range limit. At the isotropic level this choice corresponds to shifting $\wp(x)$ to $-\rho'(x)$ [22,34]; this shift also simplifies the dispersion and Bethe equations, and allows the latter to be recast in rational form [23].

**Algebraic structure at short range.** The operators $e_i \equiv S^{\mathrm{H}}_{[i,i+1]}$ in (24) obey the Temperley–Lieb (TL) relations

$$e_i^2 = 2\cos(\pi\gamma)\, e_i\,, \quad 1 \leqslant i \leqslant N-1\,, \qquad e_i\, e_{i\pm1}\, e_i = e_i\,, \quad 1 \leqslant i \leqslant N-2\,. \tag{28}$$

The boundary term (26) is a 'braid translation' [44], and $e_0 \equiv S^{\mathrm{H}}_{[1,N]}$ obeys the *periodic* TL relations, i.e. the preceding extended to subscripts mod $N$. The translation $u \propto G^{\mathrm{H}}$ enhances this to the *affine* TL algebra,

$$u\, e_i\, u^{-1} = e_{i-1\,\mathrm{mod}\,N}\,, \quad 1 \leqslant i \leqslant N\,, \qquad u^N \text{ is central}\,, \qquad u^2 e_1 \cdots e_{N-1} = e_{N-1}\,. \tag{29}$$

Thus, (24) is a dynamical alternative to the twisted Heisenberg chain of [45], relating to the affine TL algebra in a similar way as the usual TL algebra underpins the Heisenberg XXZ chain with special open boundaries [46]. Also note that (24) resembles an unrestricted version of the RSOS model [47]. It provides an $S^z$-symmetric alternative to the TL representation from the conclusion of [48], enabled by the dynamical nature of our $e_i$, cf. [49].

# 3 The quantum many body system

## 3.1 Hamiltonians

Now consider $N$ spin-$\frac{1}{2}$ particles with coordinates $x_j$ moving on a circle. Given another parameter $\epsilon$, consider the shift operator

$$\Gamma_i = \exp\!\left(-\mathrm{i}\hbar\,\epsilon\,\partial_{x_i}\right)\,, \quad x_k \mapsto x_k - \mathrm{i}\hbar\,\epsilon\,\delta_{jk}\,. \tag{30}$$

---

[3] This is supported by the fact that the principal grading operator is essential in the construction of the universal elliptic $R$-matrix of vertex type [42]. We thank H. Konno for pointing this out.

Our QMBS is given by a tower of conserved charges that are difference operators built from (30) and the deformed permutation (9). The first conserved charge is

$$
\widetilde{D}_1 = \sum_{i=1}^{N} A_i(\boldsymbol{x}) \times \quad \begin{array}{c} \text{(diagram)} \end{array} \qquad \begin{array}{c} x_i \\ \epsilon \bullet \\ x_i^- \end{array} = \Gamma_i \,, \qquad x_i^- \equiv x_i - i\hbar\epsilon \tag{31}
$$

$$
= \sum_{i=1}^{N} A_i(\boldsymbol{x}) \, P_{i-1,i}(x_i - x_{i-1}) \cdots P_{12}(x_i - x_1) \, \Gamma_i \, P_{12}(x_1 - x_i) \cdots P_{i-1,i}(x_{i-1} - x_i)
$$

$$
= \sum_{i=1}^{N} A_i(\boldsymbol{x}) \, P_{i-1,i}(x_i - x_{i-1}) \cdots P_{12}(x_i - x_1) \, P_{12}(x_1 - x_i^-) \cdots P_{i-1,i}(x_{i-1} - x_i^-) \, \Gamma_i \,,
$$

with coefficients

$$
A_i(\boldsymbol{x}) = \prod_{j(\neq i)}^{N} \frac{\theta(x_i - x_j + \eta)}{\theta(x_i - x_j)} \,. \tag{32}
$$

We furthermore have an 'antichiral' version of (31),

$$
\widetilde{D}_{-1} = \sum_{i=1}^{N} A_i(-\boldsymbol{x}) \times \quad \begin{array}{c} \text{(diagram)} \end{array} \qquad x_i^+ \equiv x_i + i\hbar\epsilon \tag{33}
$$

$$
= \sum_{i=1}^{N} A_i(-\boldsymbol{x}) \, P_{i,i+1}(x_{i+1} - x_i) \cdots P_{N-1,N}(x_N - x_i) \, \Gamma_i^{-1} \, P_{N-1,N}(x_i - x_N) \cdots P_{i,i+1}(x_i - x_{i+1})
$$

$$
= \sum_{i=1}^{N} A_i(-\boldsymbol{x}) \, P_{i,i+1}(x_{i+1} - x_i) \cdots P_{N-1,N}(x_N - x_i) \, P_{N-1,N}(x_i^+ - x_N) \cdots P_{i,i+1}(x_i^+ - x_{i+1}) \, \Gamma_i^{-1} \,.
$$

These two operators commute with each other, and with the total shift operator

$$
\widetilde{D}_N = \Gamma_1 \cdots \Gamma_N \,. \tag{34}
$$

In Section 3.4 we will describe how the higher conserved charges, whose structure is like in [28, 43, 50], are constructed.

**Example.** For $N = 3$ we have

$$
\begin{aligned}
\widetilde{D}_1 = {} & A_1(\boldsymbol{x}) \Gamma_1 + A_2(\boldsymbol{x}) P_{12}(x_2 - x_1) \Gamma_2 P_{12}(x_1 - x_2) \\
& + A_3(\boldsymbol{x}) P_{23}(x_3 - x_2) P_{12}(x_3 - x_1) \Gamma_3 P_{12}(x_1 - x_3) P_{23}(x_2 - x_3) \,, \\
\widetilde{D}_{-1} = {} & A_3(-\boldsymbol{x}) \Gamma_3^{-1} + A_2(-\boldsymbol{x}) P_{23}(x_3 - x_2) \Gamma_2^{-1} P_{23}(x_2 - x_3) \\
& + A_1(-\boldsymbol{x}) P_{12}(x_2 - x_1) P_{23}(x_3 - x_1) \Gamma_1^{-1} P_{23}(x_1 - x_3) P_{12}(x_1 - x_2) \,.
\end{aligned} \tag{35}
$$

## 3.2  Properties and limits

Our QMBS, of which (31) and (33) are the first two commuting charges, depends on the four parameters of our spin chain, as well as on the shift $\epsilon$.

239 **Defining properties.**   As $\eta \to 0$, again with $a \to -\mathrm{i}\infty$, we get the ('effective' form of the)
240 elliptic spin-CS system [51, 52].   Next, $\kappa \to 0$ and $a \to -\mathrm{i}\infty$ readily yields the spin-RM
241 system [28,50] underlying the deformed HS chain [26,28,30]. See Fig. 2. Replacing $P(x) \rightsquigarrow 1$
242 gives the spinless elliptic Ruijsenaars system [53].

243     Moreover, our QMBS is integrable in the sense that the difference operators all commute,
244 e.g.

$$[\widetilde{D}_1, \widetilde{D}_{-1}] = 0, \quad [\widetilde{D}_{\pm 1}, \widetilde{D}_N] = 0. \tag{36}$$

245 The second equality is clear as $\widetilde{D}_{\pm 1}$ only depend on coordinate differences. The first one can
246 be checked explicitly for low $N$.

## 247   3.3   Discussion

248 **Commutativity.**   It seems difficult to use the proof of integrability of [43], which relies heav-
249 ily on the periodicity properties of $\check{R}^{8\mathrm{v}}$ for simplifying expressions and setting up a proof by
250 induction. Alas, (8) does not have such simple properties. Our proof of (36) is independent.
251 In view of its technical nature it will appear elsewhere.

252 **Choice of $R$-matrix.**   Since (31)–(33) only differ from the spin-Ruijsenaars model found by
253 Matushko and Zotov [43] in the choice of $R$-matrix, (27) might again lead one to expect these
254 QMBSs to be equivalent. But, because the face-vertex transformation (27) depends on coor-
255 dinates $x_k$, it does not commute with the shift operators $\Gamma_i$. Thus our difference operators are
256 *not* face-vertex transforms of those of MZ, and define *another* QMBS. As we have seen, this
257 difference persists to all limiting spin chains (see [38] for more).

258 **Modular family.**   A new feature of the elliptic case is that there is an $SL(2,\mathbb{Z})$-family of clas-
259 sical equilibria of (49) related by modular transformations of the quasiperiods $N, \mathrm{i}\pi/\kappa$ [54].
260 These equilibria can be identified by reparametrising $\eta, a, \epsilon, \boldsymbol{x}$. Upon freezing, however, each
261 equilibrium yields a *different* integrable spin chain.

## 262   3.4   Heuristic derivation of the QMBS

263 Let us motivate how the expressions (31), (33) and (34) for the charges of our QMBS with
264 spins can be 'derived' from the *spinless* QMBS known as the elliptic Ruijsenaars system. The
265 latter describes $N$ scalar particles moving on a circle with coordinates $x_k$ and is defined by the
266 difference operator

$$D_1 = \sum_{i=1}^{N} A_i(\boldsymbol{x})\, \Gamma_i, \qquad A_i(\boldsymbol{x}) = \prod_{j(\neq i)}^{N} \frac{\theta(x_i - x_j + \eta)}{\theta(x_i - x_j)}. \tag{37}$$

267 The operator $D_1$ belongs to a hierarchy of conserved charges, i.e. commuting difference op-
268 erators. While this commutativity holds in general, it is physically reasonable to focus on
269 bosonic/fermionic wave functions with definite (anti)symmetry

$$s_{i,i+1}\Psi(\boldsymbol{x}) = \pm\Psi(\boldsymbol{x}), \qquad 1 \leqslant i < N. \tag{38}$$

270 The space of either type of wave functions is preserved by (37). At the same time, on either
271 space, (37) is determined by any single term: if we have an operator of the form $\sum_i B_i(\boldsymbol{x})\Gamma_i$
272 where, say, $B_1(\boldsymbol{x}) = A_1(\boldsymbol{x})$ is as in (37), then the prescribed symmetry fixes the remain-
273 ing coefficients to be as in (37) too. Indeed, on any wave function obeying (38) we have
274 $D_1\Psi(\boldsymbol{x}) = (\pm s_{12})D_1(\pm s_{12})\Psi(\boldsymbol{x}) = s_{12}D_1 s_{12}\Psi(\boldsymbol{x})$ since $D_1\Psi(\boldsymbol{x})$ also obeys (38); comparing
275 coefficients of $\Gamma_2$ in $D_1 = s_{12}D_1 s_{12}$ gives $B_2(\boldsymbol{x}) = s_{12}B_1(\boldsymbol{x})s_{12} = A_2(\boldsymbol{x})$. Likewise, equating

coefficients of $\Gamma_3$ in $D_1 = s_{23} D_1 s_{23}$ yields $B_3(\boldsymbol{x}) = A_3(\boldsymbol{x})$, and so on. This argument provides a useful heuristic to understand the structure of Ruijsenaars operators in more complicated settings, such as the trigonometric spin-Ruijsenaars–Macdonald system [28], the trigonometric and elliptic spin-Ruijsenaars systems of Matushko and Zotov [43], and ours.

Now consider a QMBS with $N$ *spin*-1/2 particles moving on a circle. To define bosons or fermions in our setting, the appropriate permutation operator for the particles is

$$P_{i,i+1}^{\text{tot}} = s_{i,i+1} P_{i,i+1}(x_i - x_{i+1}), \tag{39}$$

which exchanges both coordinates, through $s_{i,i+1}$, as well as spins, through $P_{i,i+1}(x_i - x_{i+1})$ as defined in (9). Such permutation operators form a representation of the braid group (see Appendix B), and reduce to the usual permutation of particles, $s_{i,i+1}P_{i,i+1}$, as $\eta \to 0$. In terms of this permutation operator the boson(fermion) condition is simply

$$P_{i,i+1}^{\text{tot}} |\Psi\rangle = \pm |\Psi\rangle, \qquad 1 \leqslant i < N. \tag{40}$$

Now suppose a difference operator has the form $\widetilde{D}_1 = \sum_i \widetilde{B}_i(\boldsymbol{x})\Gamma_i$ on either space, and again $\widetilde{B}_1(\boldsymbol{x}) = A_1(\boldsymbol{x})$. The coefficient of $\Gamma_2$ in $\widetilde{D}_1 = P_{12}^{\text{tot}} \widetilde{D}_1 P_{12}^{\text{tot}}$ can be found by comparing

$$
\begin{aligned}
\widetilde{B}_2(\boldsymbol{x})\Gamma_2 = P_{12}^{\text{tot}} \widetilde{B}_1(\boldsymbol{x})\,\Gamma_1 P_{12}^{\text{tot}} &= s_{12} P_{12}(x_1-x_2) A_1(\boldsymbol{x})\Gamma_1 s_{12} P_{12}(x_1-x_2) \\
&= A_2(\boldsymbol{x}) P_{12}(x_2-x_1)\Gamma_2 P_{12}(x_1-x_2) \\
&= A_2(\boldsymbol{x}) P_{12}(x_2-x_1) P_{12}(x_1-x_2+\mathrm{i}\hbar\epsilon)\Gamma_2,
\end{aligned}
\tag{41}
$$

whence $\widetilde{B}_2(\boldsymbol{x}) = A_2(\boldsymbol{x}) P_{12}(x_2-x_1) P_{12}(x_1-x_2+\mathrm{i}\hbar\epsilon)$. Similarly,

$$
\begin{aligned}
\widetilde{B}_3(\boldsymbol{x})\Gamma_3 &= P_{23}^{\text{tot}} \widetilde{B}_2(\boldsymbol{x})\,\Gamma_2 P_{23}^{\text{tot}} \\
&= s_{23} P_{23}(x_2-x_3) A_2(\boldsymbol{x}) P_{12}(x_2-x_1)\Gamma_2 P_{12}(x_1-x_2) s_{23} P_{23}(x_2-x_3) \\
&= A_3(\boldsymbol{x}) P_{23}(x_3-x_2) P_{12}(x_3-x_1)\Gamma_3 P_{12}(x_1-x_3) P_{23}(x_2-x_3) \\
&= A_3(\boldsymbol{x}) P_{23}(x_3-x_2) P_{12}(x_3-x_1) P_{12}(x_1-x_3+\mathrm{i}\hbar\epsilon) P_{23}(x_2-x_3+\mathrm{i}\hbar\epsilon)\Gamma_3,
\end{aligned}
\tag{42}
$$

and so on. In this way we obtain our first difference operator (31).

Its 'antichiral' counterpart $\widetilde{D}_{-1} = \sum_i \widetilde{B}_{-i}(\boldsymbol{x})\Gamma_i^{-1}$ is likewise fixed by (40) starting from the coefficient $\widetilde{B}_{-N}(\boldsymbol{x}) = A_N(-\boldsymbol{x})$ and yields (33).

More generally, the higher conserved charges $\widetilde{D}_{\pm r} = \sum_{i_1 < \cdots < i_r}^{N} \widetilde{B}_{\pm i_1, \ldots, \pm i_r}(\boldsymbol{x})\Gamma_{i_1}^{\pm 1} \cdots \Gamma_{i_r}^{\pm 1}$ are obtained in the same way from $\widetilde{B}_{1\ldots r}(\boldsymbol{x}) = A_{1\ldots r}(\boldsymbol{x}) = \prod_{i(\leqslant r)} \prod_{j(>r)}^{N} \theta(x_i - x_j + \eta)/\theta(x_i - x_j)$ and $\widetilde{B}_{-(N-r+1),\ldots,-N}(\boldsymbol{x}) = A_{N-r+1,\ldots N}(-\boldsymbol{x})$, yielding a tower of hamiltonians, with structure like in [28, 43, 50]. In particular, the total shift operator takes the simple form $\widetilde{D}_N = \Gamma_1 \cdots \Gamma_N$.

We emphasise that while this argument 'explains' the structure of our dynamical spin-Ruijsenaars operators, including the appearance of $R$-matrices, and shows that our operators preserve the 'physical space' of bosonic/fermionic vectors (40), it does *not* prove their commutativity (36). The proof will be published elsewhere in view of its technical nature.

## 3.5  Freezing

Let us discuss the relation between the spin-chain hamiltonians (4) and the spin-Ruijsenaars operators (31)–(33). We begin with a useful heuristics for deriving the spin-chain hamiltonians from the QMBS. Let $\delta = \partial_\epsilon\big|_{\epsilon=0}$ denote linearisation in $\epsilon$. Using $\delta\Gamma_j = -\mathrm{i}\hbar\,\partial_{x_j}$ and the Leibniz rule we compute

$$\delta\widetilde{D}_1 = \sum_{j=1}^{N} A_j(\boldsymbol{x}) \times \delta \quad \text{(diagram)}$$

$$= \sum_{j=1}^{N} A_j(\boldsymbol{x}) \times -\mathrm{i}\hbar \left( \partial_{x_j} - \sum_{i=1}^{j-1} \text{(diagram)} \right),$$

(43)

where the ⊛ denotes a derivative of the (deformed) permutation (9). Note that the spin and differential part decouple ('spin-charge separation'). By unitarity and recognising (10) the spin part is

$$\text{(diagram)} = \text{(diagram)}$$

$$= \theta(\eta) V(x_i - x_j) \times \text{(diagram)},$$

(44)

which equals $\theta(\eta) V(i-j) S^{\mathrm{L}}_{[i,j]}$ at $x_k^\star = k$ ($1 \leqslant k \leqslant N$). The computation of $\delta\widetilde{D}_{-1}$ is analogous, instead yielding $\theta(\eta) V(i-j) S^{\mathrm{R}}_{[i,j]}$. As we will explain below, at the equispaced positions $x_k^\star = k$ the coefficients $A_j(\boldsymbol{x}^\star) = A^\star$ have a common value $[A^\star = \theta(\eta)_{N=1}/N\,\theta(\eta)]$. Then we can conclude that

$$\frac{1}{\mathrm{i}\hbar\,\theta(\eta)} \left[ \delta\widetilde{D}_{\pm 1} \mp \sum_{j=1}^{N} A_j(\pm\boldsymbol{x})\,\delta\,\Gamma_j \right]_{x_k = x_k^\star} = \frac{1}{\mathrm{i}\hbar\,\theta(\eta)} \left[ \delta\widetilde{D}_{\pm 1} \mp A^\star\,\delta\widetilde{D}_N \right]_{x_k = x_k^\star}$$

$$= A^\star \sum_{i<j}^{N} V(i-j) S^{\mathrm{L,R}}_{[i,j]} = A^\star H^{\mathrm{L,R}}.$$

(45)

The physical picture is that $\epsilon = \mathrm{i}\,\eta/g$ (cf. the 'nonrelativistic limit' to the spin-Calogero–Sutherland system) and in the classical/strong-coupling limit $\hbar\epsilon \propto \hbar/g \to 0$ the kinetic energy is negligible compared to the potential energy, and the particles slow down to come to a halt, 'freezing' at the classical equilibrium positions $x_k^\star = k$ of the spinless elliptic Ruijsenaars system.

The expansion (45) gives the correct spin-chain hamiltonian, but the calculation has to be made more precise to turn it into a proper derivation. Here we outline how this goes; details will be given in [39]. Let us for a moment keep the elliptic parameter $\tau$ arbitrary by replacing the (odd) Jacobi theta function (5) by

$$\vartheta(x\,|\,\tau) = \frac{\sin(\pi x)}{\pi} \prod_{n=1}^{\infty} \frac{\sin[\pi(n\tau+x)]\sin[\pi(n\tau-x)]}{\sin^2(\pi n\tau)}.$$

(46)

Consider the classical spinless elliptic Ruijsenaars system with canonically conjugate coordinates $x_i$ and momenta $p_j$, with Poisson brackets $\{x_i, p_j\} = \delta_{ij}$. The ('chiral') hamiltonians are

$$D_{\pm 1}^{\text{cl}} = \sum_{i=1}^N e^{\pm \epsilon p_i} A_i(\pm \boldsymbol{x}; \eta \,|\, \tau), \qquad A_i(\boldsymbol{x}; \eta \,|\, \tau) = \prod_{j(\neq i)}^N \frac{\vartheta(x_i - x_j + \eta \,|\, \tau)}{\vartheta(x_i - x_j \,|\, \tau)}. \tag{47}$$

These functions belong to a family of $N$ independent Poisson-commuting quantities, which are the conserved charges of the classical Ruijsenaars–Schneider system [55]. Picking $D_1^{\text{cl}}$ as hamiltonian defines a time flow with velocities

$$\frac{\partial x_j}{\partial t} \equiv \{x_j, D_1^{\text{cl}}\} = \frac{\partial D_1^{\text{cl}}}{\partial p_j} = \epsilon \, e^{\epsilon p_j} A_j(\boldsymbol{x}; \eta \,|\, \tau), \tag{48a}$$

and momenta changing as

$$\frac{\partial p_j}{\partial t} \equiv \{p_j, D_1^{\text{cl}}\} = -\frac{\partial D_1^{\text{cl}}}{\partial x_j} = -\sum_{i=1}^N e^{\epsilon p_i} \partial_{x_j} A_i(\boldsymbol{x}; \eta \,|\, \tau). \tag{48b}$$

We can search for phase-space configurations $(\boldsymbol{x}^\star, \boldsymbol{p}^\star) \in \mathbb{C}^{2N}$ that satisfy the classical equilibrium conditions

$$\frac{\partial x_j}{\partial t} = \epsilon A^\star, \qquad \frac{\partial p_j}{\partial t} = 0, \tag{49}$$

for a ($j$-independent) constant $A^\star$. Such configurations are 'frozen' in the sense that they remain stationary in the co-moving frame with velocity $A^\star$. Evaluating our quantum spin-Ruijsenaars system at such stationary configurations and dropping all derivatives in a consistent manner yields a spin-chain hamiltonian like in (45), cf. [37].

One equilibrium configuration solving (49) is

$$x_j^\star = \frac{j}{N}, \quad p_j^\star = 0, \quad \tau = \frac{\omega}{N}, \tag{50}$$

(we parametrise $\omega = i\pi/\kappa$). In this case all coefficients $A_j\big(\boldsymbol{x}^\star; \frac{\eta}{N} \,\big|\, \frac{\omega}{N}\big)$ are equal to the constant $A^\star \equiv \vartheta(\eta \,|\, \omega)/\big[N \,\vartheta\big(\frac{\eta}{N} \,\big|\, \frac{\omega}{N}\big)\big]$. This configuration is used to obtain an integrable spin chain by freezing for the HS and deformed HS chains [28] and was used by Matushko and Zotov [37]. In this case the argument around (45) can be made rigorous following [37].

However, the resulting spin chain does not admit a Heisenberg-type short-range limit. Happily, there are many more solutions to (49), each belonging to a (lattice) parameter $\tau$ [39]. The modular action of $SL(2, \mathbb{Z})$ on $\tau$ relates these solutions. In particular, one of the other equilibrium configurations is

$$x_j^\star = \frac{-j}{\omega}, \quad p_j^\star = \frac{i\pi \eta}{\omega \epsilon}(N - 2j + 1), \quad \tau^\star = \frac{-N}{\omega}, \tag{51}$$

which yields the theta function (5) as $\theta(x) = \omega \, \vartheta\big(\frac{x}{\omega} \,\big|\, \frac{-N}{\omega}\big)$. Note that the positions in (51) are still equally spaced, albeit now along the imaginary axis. The values of the momenta in (51) compensate for the differences between

$$A_j\Big(\boldsymbol{x}^\star; \frac{-\eta}{\omega} \,\Big|\, \frac{-N}{\omega}\Big) = e^{-(N-2j+1)\eta \kappa} \, \vartheta\big(\tfrac{\eta}{\omega} \,\big|\, \tfrac{-1}{\omega}\big)/\vartheta\big(\tfrac{\eta}{\omega} \,\big|\, \tfrac{-N}{\omega}\big), \tag{52}$$

so that all velocities (49) are again equal; one may think of the particles as having different masses. Thus, the expansion leading to (45) has to be computed more carefully, taking into account that $\Gamma_i = e^{\epsilon \hat{p}_i} \to e^{\epsilon p_i}$ also contributes to the value of $A^\star = \vartheta\big(\frac{\eta}{\omega} \,\big|\, \frac{-1}{\omega}\big)/\vartheta\big(\frac{\eta}{\omega} \,\big|\, \frac{-N}{\omega}\big)$; see [39] for details. The result is that freezing the quantum spin-Ruijsenaars system at (51)

yields our spin-chain hamiltonians (4) with theta functions (5). Unlike the spin chain obtained by freezing at (50), this spin chain admits a short-range limit, as discussed above.

Note that (45) does not yet imply the commutativity (18) of the commuting charges of our spin chain. This can be proven [39] following [26, 32, 37] using the commutativity (36) for the spin-Ruijsenaars system. The conclusion is that the commutativity of the hamiltonians of our QMBS implies that for the hamiltonians of our spin chain.

# 4  Conclusion

**Summary.**    We introduced a new integrable long-range quantum spin chain that unifies the Inozemtsev chain and the deformed Haldane–Shastry chain: the deformed Inozemtsev chain. It is obtained by 'freezing' a quantum many body system (QMBS) of particles with spins *moving* on a circle: the dynamical elliptic spin-Ruijsenaars system, which is also new. Both models are (quantum) integrable in the sense that they possess a family of conserved charges including the hamiltonians. The freezing procedure guarantees that the commutativity of these conserved charges is preserved when passing from the QMBS to the spin chain.

Since the $SU(2)$-symmetric Inozemtsev chain is a limit of our $U(1)$-symmetric generalisation, through our work the Inozemtsev chain, too, is embedded in the framework of freezing at last. It thus gives strong evidence for its integrability (existence of many conserved charges), although extracting explicit conserved charges from (4) requires effort, cf. Remark ii in §1.3.4 of [28]. Moreover, our work provides a first glimpse of underlying algebraic structures via the appearance of $R$-matrices. The latter depend on an extra 'dynamical' parameter, not unlike suggestions of [17, 22]. Thus, our work presents a major step towards a general theory of (quantum) integrability for long-range models with spins.

Our models differ from those of Matushko and Zotov [37, 43] in that the deformed spin interactions are built from the (face-type) dynamical elliptic $R$-matrix, rather than the (vertex-type) elliptic $R$-matrix of Baxter. Unlike for periodic nearest-neighbour chains, the two sets of models are not related by a face-vertex transformation. The difference has significant implications for the physical properties, even in all limits [38].

In addition to recovering known limits, we showed that the deformed Inozemtsev chain also has two new limits. Its short-range limit is a twisted Heisenberg XXZ chain that seems to be new and is related to the affine Temperley–Lieb algebra in the spirit of [48], certainly warranting further investigation. Other promising directions are RSOS specialisations, cf. [47]. It would also be worth investigating our novel intermediate generalisation of the Inozemtsev spin chain depending on an extra parameter $a'$, which sits somewhere between the latter and its deformed generalisation in Fig. 1. The fact that the parameter $a'$ disappears in all limits (including infinite length) makes this model rather unique, and its solution structure could shed light on the particular challenges that appear at the elliptic level.

**Outlook.**    Our work opens up many new directions.

The exact characterisation of the energies and eigenstates of our models is left for future work. The spin chain magnons, eigenstates of the (twisted) translation operator, already exhibit rich structure, making it quite non-trivial to find the dispersion relation. The eigenstates of both the isotropic Inozemtsev and deformed Haldane–Shastry chain rely on a connection to a *scalar* QMBS. It is natural to investigate whether our freezing procedure can produce eigenstates for the chain from the eigenfunctions of the scalar elliptic Ruijsenaars model [56–59] as well, connecting it to elliptic Macdonald theory and elliptic toroidal algebras beyond $\mathfrak{gl}_1$, cf. [60]. Through suitable short-range limits, we believe this will provide a new perspective even on the well-known Bethe-ansatz solution of the isotropic Heisenberg chain.

396  The anisotropy of our deformed Inozemtsev chain can be set to points of special interest
397  for condensed-matter theory, where it will simplify to yield new long-range models with e.g.
398  free fermions or supersymmetry on the lattice, cf. [61].

399  Our work also has implications for high-energy theory: long-range spin chains naturally
400  appear in AdS/CFT integrability (see [62–65] and references therein), and our QMBS is closely
401  related to supersymmetric gauge theories in five dimensions, cf. [60,66,67]. Finally, it provides
402  a test for the conjectured spin-version of the (quantum) 'DELL' (double elliptic) system [66,67].

## 403  Acknowledgements

404  First and foremost we thank G. Arutyunov for introducing us to the Inozemtsev chain and
405  supporting us in the work that eventually led to this work, cf. [68]. We thank O. Chalykh,
406  G. Felder, F. Göhmann, P. Koroteev, M. Ren, H. Rosengren, D. Serban and M. Volk for interest
407  and discussions, H. Konno for correspondence, and J.-S. Caux, D. Serban, A. Sfondrini and
408  especially B. Doyon for feedback on drafts. We thank the organisers of the conference *Integra-*
409  *bility, Dualities and Deformations* at Humboldt-Universität zu Berlin (2022), where a key step
410  of this work was made. JL presented this work at *Integrability in Gauge and String Theory*,
411  ETH Zürich (2023).

412  **Funding information.** The work of JL was funded by Labex Mathématique Hadamard (LMH),
413  and in the final stage by ERC-2021-CoG – BrokenSymmetries 101044226.

## 414  A  Elliptic functions

415  Here we summarise the definitions of the elliptic functions that we need. See [23] (where
416  the functions $\theta$ and $\rho$ defined below were decorated with a subscript '2') and [39] for more
417  details or the standard references [69–71].

418  We use the (odd) Jacobi theta function with nome $p = \mathrm{e}^{-N\kappa}$, which is a periodisation of a
419  hyperbolic sine:

$$\theta(x) = \frac{\sinh(\kappa\,x)}{\kappa} \prod_{n=1}^{\infty} \frac{\sinh[\kappa\,(N\,n + x)]\sinh[\kappa\,(N\,n - x)]}{\sinh^2(N\kappa\,n)} = \frac{\sinh(\kappa\,x)}{\kappa} + O(p^2). \qquad \text{(A.1)}$$

420  It is the unique odd entire function with double quasiperiodicity

$$\theta(x + \mathrm{i}\pi/\kappa) = -\theta(x) \qquad \theta(x + N) = -\mathrm{e}^{\kappa(2x+N)}\,\theta(x) \qquad \text{(A.2)}$$

421  and normalisation $\theta'(0) = 1$. In terms of the Weierstraß sigma function with quasiperiods $N$
422  and $\mathrm{i}\pi/\kappa$ it reads

$$\theta(x) = \mathrm{e}^{\mathrm{i}\kappa\,\eta_2 x^2/2\pi}\,\sigma(x), \qquad \eta_2 = 2\,\zeta(\mathrm{i}\pi/2\kappa). \qquad \text{(A.3)}$$

423  It obeys the addition formula

$$\theta(x + y)\,\theta(x - y)\,\theta(z + w)\,\theta(z - w) = \theta(x + z)\,\theta(x - z)\,\theta(y + w)\,\theta(y - w)$$
$$+ \theta(x + w)\,\theta(x - w)\,\theta(y + z)\,\theta(y - z). \qquad \text{(A.4)}$$

424  The prepotential is the logarithmic derivative

$$\rho(x) = \frac{\theta'(x)}{\theta(x)} = \zeta(x) + \frac{\mathrm{i}\kappa\,\eta_2}{\pi}\,x = \kappa\coth(\kappa\,x) + O(p^2), \qquad \text{(A.5)}$$

with $\zeta(x) = \sigma'(x)/\sigma(x)$ the Weierstraß zeta function. It is odd and obeys $\rho(x+\mathrm{i}\pi/\kappa) = \rho(x)$, $\rho(x + N) = \rho(x) + 2\kappa$.

Finally, the potential is defined as the symmetric difference quotient

$$V(x) = -\frac{\rho(x + \eta) - \rho(x - \eta)}{\theta(2\eta)} = \frac{A}{\mathrm{sn}[B(x + \eta), k]\,\mathrm{sn}[B(x - \eta), k]} + C\,,$$
$$k = \frac{\sqrt{\wp(\mathrm{i}\pi/2\kappa) - \wp[(N + \mathrm{i}\pi/\kappa)/2]}}{\sqrt{\wp(N/2) - \wp[(N + \mathrm{i}\pi/\kappa)/2]}}\,, \tag{A.6}$$

where the equality with Jacobi's elliptic sine $\mathrm{sn}(x, k)$, with elliptic modulus $k$, involves constants $A, C$ (determined by the values at $x = 0, N/2$) and $B = \sqrt{\wp(N/2) - \wp(N/2 + \mathrm{i}\pi/2\kappa)}$. The potential is even and doubly periodic, $V(x + \mathrm{i}\pi/\kappa) = V(x + N) = V(x)$. The sign in (A.6) is chosen such that $V(x) \to -\rho'(x) = \wp(x) - \mathrm{i}\kappa\eta_2/\pi$ becomes the Weierstraß elliptic function as $\eta \to 0$.

# B  Deformed permutations

One way to obtain the dynamical $R$-matrix (8) is from Baxter's $R$-matrix of the eight-vertex model using the face-vertex transformation (27) [41, 72, 73]. As the name of the transformation suggests, one often thinks of $\check{R}(x, a)$ as defining a '(interaction-round-the-)face' (or 'IRF') model. One can equivalently view this model as a 'height model', in which case it is often called the ('elliptic' or 'eight-vertex') 'solid-on-solid' (or 'SOS') model, which can be described as a version of the six-vertex model where each face is decorated by a 'height'.

One face of the lattice is given a 'reference' height $a$, which determines the heights of all other faces by the spin configuration on the lines of the vertex model through the rule

$$a \overset{s}{\underset{s}{\uparrow}} b\,, \qquad b = a - s\,, \tag{B.1}$$

where the line carries a spin $s = \pm 1$, and $|+1\rangle \equiv |\uparrow\rangle$ and $|-1\rangle \equiv |\downarrow\rangle$. The matrix entries of the identity correspond to

$$\delta_{s,t} = \langle t|s\rangle = a \overset{t}{\underset{s}{\uparrow}} b\,, \qquad b = a - s = a - t\,, \tag{B.2}$$

Furthermore giving each line a spectral parameter, the generalised vertex model has vertices

$$\langle t', t''|\check{R}(x' - x'', a)|s', s''\rangle = a \overset{x'', t' \quad x', t''}{\underset{x', s' \quad x'', s''}{\bowtie}} c\,, \qquad \begin{aligned} b &= a - t'\,, \\ d &= a - s'\,, \end{aligned} \quad c = b - t'' = d - s''\,, \tag{B.3}$$

with (statistical-mechanical) weight equal to the corresponding entry of (8). The equality on the right uses the ice rule (spin-$z$ conservation) $s' + s'' = t' + t''$ of the dynamical $R$-matrix. By passing to the dual lattice, where the heights are instead attached to the vertices, one arrives at the standard IRF picture shown in gray in (B.1)–(B.3), with weight $W\left(a \begin{smallmatrix} b \\ d \end{smallmatrix} c \middle| x' - x''\right)$. One of the benefits of the generalised-vertex perspective is that the $R$-matrix with entries (B.3) is

just a $4 \times 4$ matrix (in the spin, rather than height, basis) as in (8), i.e.

$$
\check{R}(x,a) = \begin{pmatrix} 1 & 0 & 0 & 0 \\ 0 & f(\eta, x, \eta a) & f(x, \eta, \eta a) & 0 \\ 0 & f(x, \eta, -\eta a) & f(\eta, x, -\eta a) & 0 \\ 0 & 0 & 0 & 1 \end{pmatrix}, \quad f(x, y, z) = \frac{\theta(x)\,\theta(y+z)}{\theta(x+y)\,\theta(z)}. \quad \text{(B.4)}
$$

The price to pay is an additional parameter, $a$, that has to be shifted in the appropriate way, determined by (B.2). The dynamical $R$-matrix obeys the unitarity relation $\check{R}(x,a)\check{R}(-x,a) = 1$ and initial condition $\check{R}(0,a) = 1$. In components, unitarity reads

$$
\langle t', t'' | \check{R}(x''-x', a)\check{R}(x'-x'', a) | s', s'' \rangle = \quad \text{(diagram)} \quad = \delta_{b,d} \times \quad \text{(diagram)} \quad = \delta_{s',t'}\,\delta_{s'',t''}\,,\,, \quad \text{(B.5)}
$$

with $b = a - s'$ and $c = b - s''$, and where in the first diagram dashed lines join heights that are to be identified, and a sum over the spins on the two internal edges (equivalently, over the heights $e$ on the internal face) is understood. In addition, (B.4) obeys the (braid-like form of the) dynamical Yang–Baxter equation (or Gervais–Neveu–Felder equation)

$$
\check{R}_{12}(x'-x'', a)\check{R}_{23}(x-x'', a-\sigma_1^z)\check{R}_{12}(x-x', a)
$$
$$
= \check{R}_{23}(x-x', a-\sigma_1^z)\check{R}_{12}(x-x'', a)\check{R}_{23}(x-x', a-\sigma_1^z). \quad \text{(B.6)}
$$

In components it reads

$$
\langle t, t', t'' | \check{R}_{12}(x'-x'', a)\check{R}_{23}(x-x'', \overbrace{a-\sigma_1^z}^{=\,g})\check{R}_{12}(x-x', a) | s, s', s'' \rangle
$$

$$
= \quad \text{(diagrams)} \quad \text{(B.7)}
$$

$$
= \langle t, t', t'' | \check{R}_{23}(x-x', \underbrace{a-\sigma_1^z}_{=\,b})\check{R}_{12}(x-x'', a)\check{R}_{23}(x-x', \underbrace{a-\sigma_1^z}_{=\,f}) | s, s', s'' \rangle,
$$

where sums over spins on the three internal lines (equivalently, over the height $g$ or $h$ of the internal face) are again understood. The resulting algebraic structure is Felder's elliptic quantum group [36].

Now consider a row of $N$ vertical lines in the generalised vertex model. The deformed permutation (9) similarly encodes the vertex

$$
\langle t_1, \ldots, t_N | P_{i,i+1}(x'-x'') | s_1, \ldots, s_N \rangle = \quad \text{(diagram)} \quad, \quad \text{(B.8)}
$$

where we omitted the spectral parameters attached to all non-crossing lines to avoid cluttering, and the heights are

$$
a_0 = a, \qquad a_j = a_{j-1} - s_j \quad (j \neq i, i+1), \qquad \begin{aligned} a_i'' &= a_{i-1} - t_i, \\ a_i' &= a_{i-1} - s_i, \end{aligned} \qquad a_{i+1} = a_i'' - t_{i+1} = a_i' - s_{i+1}. \quad \text{(B.9)}
$$

The vertex (B.8) corresponds to a single matrix entry of $P_{i,i+1}(x)$. The whole matrix can be written as in (9), i.e.

$$P_{i,i+1}(x) = \check{R}_{i,i+1}\big(x, a - (\sigma_1^z + \cdots + \sigma_{i-1}^z)\big) \tag{B.10}$$

On the usual spin ('computational') basis this notation means

$$P_{i,i+1}(x)|s_1,\ldots,s_N\rangle = |s_1,\ldots,s_{i-1}\rangle \otimes \big(\check{R}\big(x, a - \textstyle\sum_{k=1}^{i-1} s_k\big)|s_i,s_{i+1}\rangle\big) \otimes |s_{i+1},\ldots,s_N\rangle. \tag{B.11}$$

We stress once more that the dynamical parameter of the $R$-matrix in (B.10)–(B.11) is shifted by (twice) the spin-$z$ to the left of the $\asymp$ in agreement with (B.8). Projecting on $\langle t_1,\ldots,t_N|$ we recover (B.8).

Thanks to (B.6), the deformed permutations obey the (braid-like) Yang–Baxter equation

$$P_{i,i+1}(x-y)\,P_{i+1,i+2}(x)\,P_{i,i+1}(y) = P_{i+1,i+2}(y)\,P_{i,i+1}(x)\,P_{i+1,i+2}(x-y), \tag{B.12}$$

as well as the commutativity $[P_{i,i+1}(x), P_{j,j+1}(y)] = 0$ for $|i-j| > 1$. They moreover inherit the unitarity relation

$$P_{i,i+1}(-x)\,P_{i,i+1}(x) = 1. \tag{B.13}$$

with 'initial condition' $P_{i,i+1}(0) = 1$. According to (B.13), swapping twice is the identity. That is, taking into account that the parameters follow the lines, the deformed permutations square (appropriately interpreted) to the identity. This can be made precise by introducing the coordinate permutation $s_{ij} : x_i \leftrightarrow x_j$. Consider the deformed total permutation

$$P_{i,i+1}^{\text{tot}} = s_{i,i+1}\,P_{i,i+1}(x_i - x_{i+1}). \tag{B.14}$$

It permutes particles, i.e. spins *and* coordinates. (Since parameters should follow lines in diagrams, one could draw it as $\vdash\!\uparrow$.) Now (B.6) becomes the braid relation

$$P_{i,i+1}^{\text{tot}}\,P_{i+1,i+2}^{\text{tot}}\,P_{i,i+1}^{\text{tot}} = P_{i+1,i+2}^{\text{tot}}\,P_{i,i+1}^{\text{tot}}\,P_{i+1,i+2}^{\text{tot}}, \tag{B.15}$$

we have $\big[P_{i,i+1}^{\text{tot}}, P_{j,j+1}^{\text{tot}}\big] = 0$ for $|i-j| > 1$, and (B.13) reads

$$\big(P_{i,i+1}^{\text{tot}}\big)^2 = 1. \tag{B.16}$$

These are the relations of the permutation group. In the isotropic limit $\eta \to 0$ we recover the standard particle permutation, $P_{i,i+1}^{\text{tot}} \to s_{i,i+1}\,P_{i,i+1}$. For general $\eta$, (B.14) depends on all parameters.

## C   Deformed nearest-neighbour exchange

The deformed spin exchange

$$E(x,a) = \frac{1}{\theta(\eta)\,V(x)}\check{R}(-x,a)\check{R}'(x,a) = \begin{array}{c} x'\ x'' \\ a\!\!\uparrow\!\!\rightsquigarrow\!\!\uparrow \\ x'\ x'' \end{array}, \quad \check{R}'(x,a) \equiv \partial_x \check{R}(x,a), \quad x = x' - x'', \tag{C.1}$$

is nothing but a normalised logarithmic derivative of the dynamical $R$-matrix, $\partial \log \check{R} = \check{R}^{-1}\check{R}'$, mirroring the local hamiltonians of Heisenberg chains. As an explicit $4 \times 4$ matrix it reads

$$\theta(\eta)\,V(x)\,E(x,a) = \check{R}(-x,a)\check{R}'(x,a) = \begin{pmatrix} 0 & 0 & 0 & 0 \\ 0 & \alpha(x,\eta\,a) & \beta(x,\eta\,a) & 0 \\ 0 & \beta(x,-\eta\,a) & \alpha(x,-\eta\,a) & 0 \\ 0 & 0 & 0 & 0 \end{pmatrix}, \tag{C.2}$$

where the first equality uses the unitarity $\check{R}(x,a)^{-1} = \check{R}(-x,a)$, and the coefficients are

$$
\begin{aligned}
\alpha(x,a) &= f(\eta,x,a)f(\eta,-x,a)\big(\rho(x+a)-\rho(x)\big)-\big(\rho(x+\eta)-\rho(x)\big) \\
&= f(\eta,x,a)f(\eta,-x,a)\rho(x+a)+f(x,\eta,a)f(-x,\eta,-a)\rho(x)-\rho(x+\eta), \quad \text{(C.3)} \\
\beta(x,a) &= f(x,\eta,a)f(\eta,-x,a)\big(\rho(x)-\rho(x-a)\big).
\end{aligned}
$$

Its entries can be interpreted like in (B.5): if '⊛' marks the derivative of $\check{R}'$,

$$
\langle t',t''|E(x'-x'',a)|s',s''\rangle = \;
\begin{array}{c} x',t' \quad x'',t'' \\ a\!\!\begin{array}{c} b \\ \wwww \\ d \end{array}\!\!c \\ x',s' \quad x'',s'' \end{array}
\; = \; \frac{1}{\theta(\eta)V(x'-x'')}\;
\begin{array}{c} x',t' \quad x'',t'' \\ a\begin{array}{c}b\\e\\a\;\otimes\;d\end{array}c \\ x',s' \quad x'',s'' \end{array}
\;, \quad
\begin{array}{l} b=a-t', \\ d=a-s', \end{array}
\quad \text{(C.4)}
$$

with $c=b-t''=d-s''$.

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
