# Peer review of "The deformed Inozemtsev spin chain"

_SciPost Physics_

## Round 1 · Referee Report · Anonymous (Referee 1) · 2024-7-29

Strengths

Formulation of a deformation the Inozemtsev model, which can be understood as the “freezing" of a corresponding quantum many-body system with spin.

Weaknesses

  1. Proofs of integrability are not provided
  2. The models are rather complicated

Report

The Haldane-Shastry model has many unusual and remarkable properties. Its discovery in 1988 spawned an effort, continuing to this day, to understand its relation to more familiar integrable models. The paper by Klabbers and Lamers is part of this on-going effort. The authors introduce a deformed Inozemtsev model, which reduces to the XXZ, deformed Haldane-Shastry, and Inozemtsev models in appropriate limits, nicely summarized in Figure 1. This model can be understood as the “freezing" of a new quantum many-body system with spin, which reduces to the trigonometric spin-Ruijsenaars-Macdonald and elliptic spin-Calogero-Sutherland models in corresponding limits, see Figure 2. The authors claim that these models are integrable, but they leave proofs for future publications.

Although the new models are rather complicated (even for the simpler Inozemtsev model, the spectrum is only partially understood), it makes sense to investigate them. I find the paper interesting and well written. I do have some suggestions for improving the presentation, as listed below.

Requested changes

  1. The word “periodization” seems to have definitions different from what the authors presumably have in mind (see e.g. https://en.wikipedia.org/wiki/Periodization ), and should be changed in order to avoid confusion, e.g. “periodic version of ‘’

  2. Add a comment on the meaning of ~ in (6)

  3. Add a comment on the notation * ( used extensively in Sec. 3.5), perhaps below (11), where it first appears.

  4. Since the limit $a \rightarrow - i \infty$ is referenced multiple times, it would be helpful to add some details on how it is computed.

  5. There is a missing equation number (??) near the bottom of page 8.

  6. Reference [54] reads “See Supplemental Material for background and details.’’ Do the authors mean the Appendices, or some additional material? If the former, then this should be stated explicitly. If the latter, then where is it? (I don’t find it either on arXiv or SciPost.)

Recommendation

Ask for minor revision

---

## Round 1 · Referee Report · Anonymous (Referee 2) · 2024-8-13

Strengths

  1. manages to solve a longstanding problem (the anisotropic version of the Inosemstev model), and implies the solvability of the isotropic limit itself

  2. clear presentation in spite of the technical complexity of the result

Weaknesses

  1. the proofs are delayed for a more technical publication

Report

The present paper deals with an integrable deformation of the Inozemtsev model. The Inozemtsev model
is a long-range version of the isotropic Heisenberg model, where N spins on a regular 1dimensional lattice with periodic boundary conditions interact with each other with a potential given by an elliptic (Weierstrass) function of the distance between them. The two periods of the Weierstrass function are given by the circumference of the circle and by an imaginary parameter which sets the range of the interaction.
In the two limits of this continuous parameter one retrieves the isotropic Heisenberg model and the isotropic Haldane-Shastry (HS) model. The model was used as a benchmark for studying long-range deformations
of spin chains and their Yangian symmetry, as well as wrapping effects when the range of the interaction exceeds the length of the spin chain.

Recently, the solution of an anisotropic (or quantum deformed) version of the HS model was studied in detail, and the present work is extending most of the anisotropic HS results
to the anisotropic Inozemtsev model. The key of the construction is the so-called Felder dynamical R-matrix,
which enters to the definition of the multi-spin interaction. Although the model does not possess translational invariance it possesses an equivalent, deformed lattice translation operator.

The paper announces the main result for the structure of the Hamiltonian and discusses several limiting cases of the four different parameters on which it depends. The proof of integrability, as well as the determination of the magnon dispersion, is left for a future publication. Proving integrability for the deformed model will imply a proof for the undeformed one, a was question which was open for several decades.
A different anisotropic deformation of the Inozemtsev model was introduced recently by Matushko and Zotov; the two models are different.

The results reported here are important for establishing the landscape of various long-range integrable models and the methods used to solve them, with potential applications both in condensed matter and high energy physics.

The paper is overall clear and well written.

In conclusion, I think the paper solves a longstanding, difficult problem in the fiels of long-range integrable model, and I fully recommend its publication after the minor changes pointed out above.

Requested changes

  1. The meaning of the sentence line 148-149 is not completely obvious to me, please clarify

  2. On line 196 there is an undefined reference.

Recommendation

Publish (surpasses expectations and criteria for this Journal; among top 10%)

---

## Round 1 · Referee Report · Anonymous (Referee 3) · 2024-9-11

Strengths

1) The paper is well written and clearly presented, a joy to read

Weaknesses

1) I doubt that the model itself has much relevance for any application in physics. In my understanding the paper is a mathematical paper which may potentially help to understand the more relevant Inozemtsev model.

2) It seems the paper was originally written as a letter and submitted to a different journal (which can be guessed from the archive version). Because of the page number limitation of that journal, important details, like the proof of commutativity of the difference operators introduced in the paper, are omitted.

Report

The authors construct a new lattice model with an interesting underlying algebraic structure that is defined in terms of the Hamiltonians $H^L$ and $H^R$, equation (4) of the manuscript. These are obtained, by a technique developed in the 90s and called freezing, from corresponding difference operators (eqs. (31)-(34)). The difference operators are part of a family of higher difference operators that are mutually commuting. The existence of this structure, whose full presentation and proof is postponed to future work, is probably the reason why the authors like to call their model integrable. The proposed Hamiltonians $H^L$ and $H^R$ generalize the Inozemtsev Hamiltonian in that the exchange interaction of the latter is replaced by an anisotropic an non-bilocal operator. This operator is costructed from R-matrices that are solutions of a dynamical Yang-Baxter equation and appear in the work of Felder and Varchenko on the 8-vertex model. The dynamical R-matrices together with coordinate shifts are also the building blocks of the difference operators that define the many-body system underlying the proposed lattice model.

The results seem plausible and are well presented. I recommend the paper for publication in SciPost.

I would have preferred a presentation of the full result, including the full construction of the difference operators and a proof of their mutual commutativity. However, I understand that it would require a too drastic change of the manuscript at this stage that is moreover waiting for publication for too long already.

Here are a number of minor questions and remarks. When the authors say "the spectrum is real" like in line 124 and in line 165, they say it because the Hamiltonian is Hermitian under the canonical scalar product, right? If yes it would be clearer (at least to me) if it would be stated like that. In lines 148 and 150 a reference (2.2) appears which I could not locate. (2.2) appears only as a subsection label, but there the reference does not make sense to me. In the last term in equation (21) a bracket seems to be missing. In line 198 Baxter may deserve a citation for his R-matrix. In the reference section names are sometimes spelled in lowercase, e.g. "ising" in [14], "heisenberg" in [16], "inozemtsev" in [67] which should be corrected before publication.

Requested changes

I would appreciate, though not require it, if the authors would use the term "integrable" somewhat less loosely throughout the manuscript. In the context of quantum systems the term is not well defined. To me it sounds funny that the authors say in line 44 "believed to be integrable" (to me a nonsensical statement) and in line 52, only slightly below, "while remaining integrable" (is the belief retained?). As the authors know well, the mere existence of operators commuting with the Hamiltonian is not a good criterion for integrability in the quantum case, as otherwise every Hermitian operator would be integrable, since it has a spectral decomposition of the form $\sum E_n P_n$, with mutually commuting orthogonal projectors $P_n$ and energy eigenvalues $E_n$. In my understanding, when we are talking about integrability in the quantum case, we should always refer to some algebraic structure like the Yang-Baxter equation or commuting difference operators, or we should be able to state that a model is unitarily equivalent to a non-interacting model, like e.g. in case of the Calogero model.

Recommendation

Publish (easily meets expectations and criteria for this Journal; among top 50%)

---

## Editorial Decision

resubmitted